# Polymerizable Matrix Metalloproteinases’ Inhibitors with Potential Application for Dental Restorations

**DOI:** 10.3390/biomedicines9040366

**Published:** 2021-03-31

**Authors:** Helena Laronha, Inês Carpinteiro, Jaime Portugal, Ana Azul, Mário Polido, Krasimira T. Petrova, Madalena Salema-Oom, Isabel Barahona, Jorge Caldeira

**Affiliations:** 1Centro de investigação interdisciplinar Egas Moniz, Instituto Universitário Egas Moniz, 2829-511 Caparica, Portugal; h.laronha@campus.fct.unl.pt (H.L.); icarpinteiro@egasmoniz.edu.pt (I.C.); aazul@egasmoniz.edu.pt (A.A.); mpolido@egasmoniz.edu.pt (M.P.); moom@egasmoniz.edu.pt (M.S.-O.); ibarahona@egasmoniz.edu.pt (I.B.); 2UCIBIO and LAQV, Requimte, Faculdade de Ciências e Tecnologia, Universidade Nova de Lisboa, 2829-516 Caparica, Portugal; k.petrova@fct.unl.pt; 3Unidade de Investigação em Ciências Orais e Biomédicas (UICOB), Faculdade de Medicina Dentária, Universidade de Lisboa, Rua Professora Teresa Ambrósio, 1600-177 Lisboa, Portugal; jaime.portugal@fmd.ulisboa.pt

**Keywords:** matrix, metalloproteinases, Inhibitors of matrix metalloproteinases, Cytotoxicity

## Abstract

Collagen cleavage by matrix metalloproteinase (MMP) is considered a major cause of dental resins long term failure. Most MMP inhibitors display significant toxicity and are unsuitable for dental resins’ applications. Here we report a study of a new class of inhibitors that display the unique property of being co-polymerizable with other vinyl compounds present in commercial dental resins, limiting their release and potential toxicity. Computational affinity towards the active site of different MMP-1; -2; -8; -9 and -13 of several compounds showed interesting properties and were synthesized. These free compounds were tested concerning their toxicity upon contact with two different cell types, with no substantial decrease in cell viability at high concentrations. Even so, compound’s safety can be further improved upon copolymerization with commercial dental resins, limiting their release.

## 1. Introduction

The matrix metalloproteinases (MMPs) belong to the Metzincs superfamily [1,2,3,4,5,6,7,8], and are characterized by a common conserved catalytic domain: VAAHExGHxxGxxH [2,5,6,7,9,10,11,12,13,14]. The MMPs are zinc dependent, functional at neutral pH [13,15] and they are a family of proteolytic enzymes with different substrates, but which share similar structural characteristics [3,5,7,8,10,11,12,13,14,15,16,17,18]. In human tissue there are 23 different types of MMPs [1,3,5,6,7,11,12,15,16] and they can be subdivided according to substrate specificity, sequential similarity and domain organization into collagenases (MMP-1, -8 and -13), gelatinases (MMPs-2 and -9), stromelysins (MMPs-3, -10 and -11), matrilysins (MMP-7 and -26), membrane-type MMPs (MMPs-14, -15, -16, -17, -24 and -25) and other MMPs (MMPs-12, -19, -20, -21, -23, -27 and -28) [1,3,6,7,8,11,12,13,14,15,16,19]. The principal biologic function of MMPs is degradation of extracellular matrix (ECM) proteins and glycoproteins, membrane receptors, cytokines and growth factors [1,2,3,5,6,7,8,11,13,14,16,20,21,22,23,24]. They are involved in several biologic processes [1,2,7,8,10,11,13,15,16,23,25,26] and their deregulation leads to the progression of various pathologies [1,2,8,13,14,19,23]. 

The MMPs are molecular targets for the development of therapeutics and diagnosis agents [14]. The inhibition of MMP activity can be done at the biomolecular expression or enzyme activity [10,16,17]. The MMPs inhibitors that could affect enzyme activity can be divided into endogenous specific or non-specific inhibitors or synthetic inhibitors [1,3,7,8,12,13,14,16,23,27,28,29,30]. The synthetic MMP inhibitor must have a functional group able to chelate the catalytic zinc ion, at least one group that promote hydrogen bonds and side chains undergoing Van der Waals interactions with the enzyme [11,13,16,31,32,33]. These requirements are crucial to the selectivity of MMP inhibitor, to increasing the efficacy and to preventing the side effects [16]. Several strategies have been suggested to create a specific MMP inhibitor [34], but they have been difficult to develop because MMPs are involved in various pathways. It is necessary to identify the enzymes that are involved in the disease progression, while there are more than 50 human metalloproteinases (23MMPs; 13 ADAM and 19 ADAMTs) [16,17].

There is a wide variety of inhibitors, but few have selectivity and specificity for MMPs [2,27,28,29,30,31,33,35] and most inhibitors have a biphenyl group conjugated to a sulfonamides group [36,37,38,39,40,41]. In this work, a set of several molecules with biphenyl group conjugated with methacrylate or methylacrylamide were studied via computational docking, of which three molecules [42] were selected to be synthesized and tested for cytotoxicity in mouse embryo NIH/3T3 fibroblast cells and human MG-63 osteoblast-like cells.

## 2. Materials and Methods

### 2.1. Computational Docking Studies

The 2D structure of the studied compounds was drawn (Figure 1) and the 3D structure of compounds was previously optimized by RHF/3–21G using Chem3D. The crystal structure of the MMPs-1, -2, -8, -9 and -13 was retrieved from the RCSB protein database (Table 1).

In SeeSAR, the binding site of these proteins was defined, containing the conserved sequence VAAHExGHxxGxxH and the S’1 pocket and the number of solutions defined was the TOP10. The results were edited by Discovery Studio and the 2D image were obtained by PoseView.

### 2.2. Synthesis of A, B and C Compounds

Synthesis was performed according to reference [42]. Briefly a diol moiety was substituted with two additional side chains to yield the final molecule.

### 2.3. Cytotoxicity Study Using MTT Assay

Two cell lines, fibroblastic cells NIH/3T3 from mouse embryo (93061524) obtained from Health Protection Agency Culture Collections and supplied by Sigma and MG-63 osteoblast-like cells from human osteosarcoma (ATCC^®^ CRL-1427^TM^) were used for cytotoxicity studies. Defrosting and handling of cell lines was carried out according to supplier information.

The NIH/3T3 cells were grown in Dulbecco’s Modified Eagle Medium (DMEM, Sigma) supplemented with 10% calf bovine serum (Sigma) and 1% Penicillin-Streptomycin solution: 10,000 U/mL penicillin, 10 mg/mL streptomycin, Sigma) and 2.5 mg/L amphotericin B (fungizone, Sigma). MG-63 cells were grown in Eagle’s Minimum Essential Medium (EMEM, Sigma) supplemented with 10% fetal bovine serum and 1% of antibiotics and 2.5 mg/L antimycotics solutions. 

Extracts from each compound A, B or C were prepared by spreading 20 μL of a solution with 20 μM, 50 μM or 100 μM of each compound in a glass petri dish, allowed to dry, and incubated at 37 °C under humidified 5% CO_2_ for 24 h with 10 mL of the corresponding growth medium according to ISO 10993–12: 2012. 

The cytotoxicity assay was performed as follows: 1 × 10^4^ cells per well, from passages 8 to 12, were seeded in 96-well plates (eight replicates) and incubated at 37 °C under humidified 5% CO_2_ for 24 h. After proliferation, the cell medium was replaced by 200 μL of each extract and incubated under the same conditions for 24 h. As positive control cells were grown in their medium containing 7.5% DMSO (Sigma) and as negative control cells proliferated in medium without any toxic corresponding to 100% cell viability. MTT assays were performed as previously described [43]. Briefly, extracts or medium were removed, an MTT solution (0.5 mg/mL prepared in serum-free medium, Sigma) was added to each well and cells were incubated for additional 3 h. After that, a solution containing 0.1% IGEPAL (Merck), 4 mM HCl (Sigma) in isopropanol (Sigma) was used to dissolve formazan. Plates were incubated with agitation for one hour and absorbance was read at 595 nm with a microplate reader (Bio-Rad^®^, Hercules, California, USA).

### 2.4. Statistical Analysis

Data analysis was performed using SPSS for MAC, 26 version (Statistic Package for Social Sciences, Inc., New York, NY, USA) and the significant level was set at 5%. Normality distribution of quantitative variables was assessed using Shapiro-Wilk test. A three-way ANOVA was performed to evaluate the effect of cell type, inhibitor compound and inhibitor concentration on cell viability. Due to the existence of interactions between the main factors, the data obtained for each type of cell were analyzed with a two-way ANOVA followed by Tukey HSD post-hoc tests (α = 0.05).

## 3. Results and Discussion

### 3.1. Docking Studies

#### 3.1.1. Physicochemical and ADMET Properties of NNGH Commercial Inhibitor, Compounds A, B and C 

Knowledge of physicochemical properties and physiological behavior as absorption, distribution, metabolism, excretion and toxicity (ADMET data) is important to predict efficacy and safety and can help drug design [44]. The SeeSAR platform provides a set of physicochemical information (Appendix A), the most relevant of which are shown in Table 2.

The molecular weight (MW) is an important parameter because it can influence several events, such as absorption, bile elimination rate and blood-brain barrier penetration [45,46,47,48,49], while hydrogen bonds acceptors and donors are important parameters for polarity and permeability [50]. Previous studies showed that, for drug development, the number of the hydrogen bonds donors may be more crucial than the number of hydrogen bonds acceptors since compounds with more hydrogen bonds acceptors have more favorable profiles related to bioavailability and membrane permeability [45,48,51,52]. According to these studies, NNGH inhibitor and compounds A and B have better bioavailability and membrane permeability than compound C. Other coefficients as partition, log P (for neutral compounds, such as compounds A, B and C) and distribution Log D (for ionizable compounds, such as NNGH inhibitor) are measures of lipophilicity a rather important characteristic that also impacts certain ADMET parameters and drug potencies. Generally, high lipophilicity compromises water solubility, being more likely to cause toxic effects, though too low lipophilicity could excessively decrease permeability and absorption [53]. Compound B, with the highest lipophilicity value (Log P = 5.089), is the least soluble compound, while the NNGH inhibitor has a low permeability because it has low lipophilicity (Log P = 1.472). These parameters belong to Lipinski’s rule of five, which determine if a compound with biological activity has physicochemical properties that would make it a likely orally active drug. All compounds respect Lipinski’s rule of five, except for compound B that exceeds the threshold Log P. More importantly, as inhibitors to be used in dentistry, will be permanently covalently bound to dental resins, decreasing drastically the eventual free concentration in the organism. 

The Topological Polar Surface Area (TPSA) is defined as the surface sum over all polar atoms and is another descriptor related to hydrogen bonding, also important to estimate the permeability and the oral bioavailability. Several models show that these properties decrease with the increase of TPSA and, for Central nervous system (CNS) permeability by passive diffusion, the TPSA must be less than 80 Å^2^ [45,50,54,55]. Compound A has the highest permeability, while the NNGH inhibitor with the highest TPSA value presents the lowest permeability and is not able to diffuse to CNS (TPSA > 80 Å^2^).

Drugs directed towards CNS cross the blood-brain barrier (BBB) by passive diffusion, or active transport mechanism [56]. The BBB penetration should be minimal in non-CNS compounds to reduce the possibility of undesired pharmacological events and neurotoxicity [50]. The most common parameter to quantify BBB penetration is the BBB log ([brain]:[blood]) and it determines the total extent of brain exposure, at a steady-state [57]. The P-glycoprotein (P-gp) is an important transporter that belongs to the ATP-binding Cassette superfamily and it can be found in cells throughout the body, including the blood-brain barrier [48,50]. The P-gp plays an important role in the distribution of drugs due to its ability to remove/extract a structurally diverse range of molecules and can reduce drug accumulation in tissues [48,58]. For the compounds A, B and C, none can cross the BBB, since they have a low BBB log value. The NNGH, for having a TPSA greater than 80Å2, is also not able to cross the BBB, despite having a high BBB log. Only compound C can be a substrate for P-gp transporters.

#### 3.1.2. Interactions between Ligand and Protein 

A relevant feature of inhibitor specificity is the way it performs as a ligand. The active site of MMPs is a deep cavity and the catalytic domain of different MMPs share a sequential similarity [59]. The catalytic domain is highly conserved, containing three histidine residues responsible for chelates the catalytic zinc ion (VAAHExGHxxGxxH) [1,14,33] and, in the terminal zone, there is the “met-turn” [1,7], that forms the outer wall of pocket S_1′_ [1,14]. There are six S pockets (S_1_, S_2_, S_3_, S_1′_, S_2′_ and S_3′_) [14] but the S_1′_ pocket is the most important since it is a determining factor for substrate specificity [1,3,11,16]. The depth of the S_11′_ pocket can be shallow (MMP-1), intermedium or deep (MMP-2, -8, -9 and -13) cavity (Figure 2) and it is highly hydrophobic [1,7,10,11,12,14,16,60]. The highly conserved sequence (VAAHExGHxxGxxH), the “met-turn” region and the S_1′_ pocket from collagenases (MMP-1, -8 and -13) and gelatinases (MMP-2 and -9) were considered for docking studies.

The catalytic activity of MMPs requires a catalytic zinc ion, a water molecule flanked by three histidine and one glutamate residues [1,8,16], present in conserved sequence- VAA**HE**xG**H**xxGxx**H**. Our hypothesis is that if there is an interaction between the compound and the catalytic zinc ion or any of these residues from the conserved sequence, there will cause enzyme inhibition. The zinc ion demonstrated affinity for all compounds, mainly for the sulfonyl group of NNGH inhibitor and for the aromatic rings of the compounds A (Figure 3a), B (Figure 3b) and C. The three histidine and glutamate residues of the conserved sequence also demonstrated capacity to establish a hydrogen bond with the hydroxamic acid of NNGH inhibitor, in MMPs-1, -2, -8 and -9 and with the methacrylate group of compounds A and B, in all MMPs or methylacrylamide group of compound C, in MMP-1, -2, -8, and -9. Another hypothesis is that the compound can cause enzyme inhibition if blocks the access to S_1′_ pocket and all compounds besides demonstrated affinity to the active site blocked totally (Figure 3d) or partially (Figure 3c) the S_1′_.

However, there were additional interactions between the compounds and the enzyme depending on the enzyme class (collagenases and gelatinases) (See Appendix B, Appendix C, Appendix D and Appendix E). Preliminary biochemical tests (not shown) gave encouraging indications on activity reduction achieved for most MMPs by compound A, B and C, even though their low solubility and polymerization tendency.

##### Collagenases (MMP-1, -8 and -13) 

The residues Leu_81_, Ala_82_ and Ala_84_ of the MMP-1, the residues Leu_160_, Ala_161_ and Ala_160_ of the MMP-8, and the residues Leu_82_, Ala_83_ and Ala_85_ of the MMP-13 are located at the same distance from the conserved sequence in the three enzymes and their neighboring peptide bond groups can establish hydrogen bonds with the compounds studied (Table 3), through the sulfonyl group or hydroxamic acid of the NNGH inhibitor, both oxygen atoms of the methacrylate group from compound A, CN group of the compounds A and B, the oxygen atom adjacent to the biphenyl group of the compound B and the methylacylamide group of the compound C.

In MMP-1, there are three more spots on the peptide that can interact with the studied compounds. The peptide bond group adjacent to Gly_79_ can establish a hydrogen bond with the NH group of methylacrylamide group of compound C (Figure 4a). The residue Asn_80_ interacts with the tertiary amine of NNGH inhibitor (Figure 4b) and with the oxygen atom adjacent to the biphenyl group of compound A (Figure 3a). The Tyr_140_ can interact with the CN group of compound B (Figure 3b) or with the oxygen atom of the methylacrylamide group of compound C (Figure 4a). In MMP-8, the residue His_162_ establishes a hydrogen bond with the oxygen atom of the carbonyl group of methacrylate function of compound B (Figure 4c). In MMP-13, the residue Thr_142_ establishes a hydrogen bond with compound C, through the oxygen atom of the methylacrylamide group (Figure 4d) and residue Thr_144_ can establish an interaction with the oxygen atom of the methacrylate group of compound B (Figure 3d).

###### Gelatinases (MMP-2 and -9)

In MMP-2, the peptide bond adjacent to Leu_83_ and Ala_84_ can establish hydrogen bonds with all of the compounds. In the case of the NNGH inhibitor, the oxygen atom of the sulfonyl group and the NH group of the hydroxamic acid interact with both residues. The oxygen atom of the carbonyl group of methacrylate in compound A can establish a hydrogen bond with Leu_83_ and Ala_84_ (Figure 5a). The CN group of compound A and B can interact with Leu_83_ or Ala_84_, in the case of compound B (Figure 5b). The oxygen atom adjacent to the biphenyl group in compound B can establish a hydrogen bond with Leu_83_ (Figure 5c). The methylacrylamide group of compound C interacts with Leu_83_ and Ala_84_ (Figure 5d). The residue His_85_ can establish a hydrogen bond with the oxygen atom of the sulfonyl group in the NNGH inhibitor and with the methacrylate group in compound B (Figure 5c).

The residues Leu_137_, Tyr_142_ and Thr_143_ belong to the “met-turn” region and interact only with the compounds A, B and C. The residue Leu_137_ establishes a hydrogen bond only with the NH group of the methylacrylamide of the compound C (Figure 5d). The residue Tyr_142_ can interact with one or both aromatic rings of the biphenyl group of the compounds A and B (Figure 5b). The residue Thr_143_ establishes a hydrogen bond with the oxygen atom of the methacrylate group, in compounds A and B or methylacrylamide group in compound C (Figure 5b,d). 

In MMP-9, the residues Leu_188_, Ala_189_ and His_190_ interact with the NNGH inhibitor and the compounds A, B and C. The sulfonyl group and the NH group of the hydroxamic acid of the NNGH inhibitor (Figure 6a) and the methacrylate group of compound A (Figure 6b) can interact with Leu_188_ and Ala_189_. In compound B, the methacrylate group establishes a hydrogen bond with His_190_ and the oxygen atom adjacent to the biphenyl group interacts with Leu_188_ and Ala_189_ (Figure 6c). The methylacrylamide group of compound C can establish a hydrogen bond with all amino acids (Figure 3c and Figure 6d). This group of compound C can also establish a hydrogen bond with the residue Asp_235_ in MMP-9, a variable residue of the conserved sequence-VAAHExGHxxG**x**xH (Figure 6d).

### 3.2. Cytotoxicity Study

In general, MG-63 cells showed significantly (*p* < 0.001) more resistance to the presence of MMP inhibitors than NIH/3T3 cells. The compound interaction with cellular membrane depend on their physical and chemicals characteristics [61]. In fact, cellular differentiation can be the answer why osteoblasts (MG-63) are more resistant than fibroblasts (NIH/3T3); it is known that growing factors and gene expression are the responsible ones for this differentiation [61,62].

Compounds A, B and C decrease cell viability (Figure 7 and Table 4) and can even be considered toxic for NIH/3T3 cells at 50 and 100 µM concentration since the number of viable cells decreased more than 30% (ISO 10993–5: 2009). The viability of NIH/3T3 cell was significantly (*p* < 0.05) higher for compound A than for compound B and C; and significantly (*p* < 0.001) higher at 20 µM than the other two concentrations tested. 

A decrease in cell viability was also observed in human MG-63 cells, but not statistical significantly differences were found between compounds (*p* = 0.307) and concentrations (*p* = 0.406). However, according to ISO standards, only compound B at 50/100 μM and compound C at 100 μM were toxic. Nevertheless, none of the three compounds highlighted serious toxicity.

Although compounds A, B and C present some toxicity towards the two cell types, it may be much less as compounds are to be used copolymerization with dental resins, which reduce their free concentration. Preliminary results by FTIR spectroscopy using commercial resins showed a decrease in the vinyl bands of the free compounds upon light polymerization (results not shown).

## 4. Conclusions

According to the computational docking simulations, all compounds (NNGH inhibitor and the compounds A, B and C) have the potential to be good inhibitors of MMPs. These compounds were chosen among a pool of previously synthesized compounds in the laboratory and a few hundred other proposals as the ones with better affinity score towards the MMPs active sites. The compounds showed an affinity for the active center of MMPs, which may block access to the S_1′_ pocket, and they can interact with the catalytic zinc ion. The residues histidine and glutamate, belong to the conserved sequence, and some residues of the “met-turn” can also establish hydrogen bonds with the compounds. Altogether available data lead us to conclude that the compounds interact with the crucial components for the catalytic activity of MMPs. Furthermore, cell viability does not decrease much upon direct contact with the free non-polymerized compounds and toxicity is expected to further decrease upon covalent binding to the dental resin matrix, promising improvements in dental resins mechanical strength.

## Figures and Tables

**Figure 1 biomedicines-09-00366-f001:**
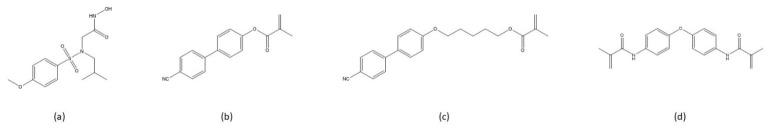
Compounds selected from docking studies. (**a**) N-Isobutyl-N-(4-methoxyphenylsulfonyl)glycyl hydroxamic acid- NNGH, inhibitor of commercial MMP enzymatic test kit (**b**) 4′-cyano-[1,1′-bihenyl]-4-yl methacrylate- Compound A [42] (**c**) 5-((4′-cyano-[1,1′-biphenyl]-4-yl)oxy)pentyl methacrylate- Compound B [42] (**d**) N,N′-(oxybis(4,1-phenylene))bis(2-methylacrylamide)- Compound C [42].

**Figure 2 biomedicines-09-00366-f002:**
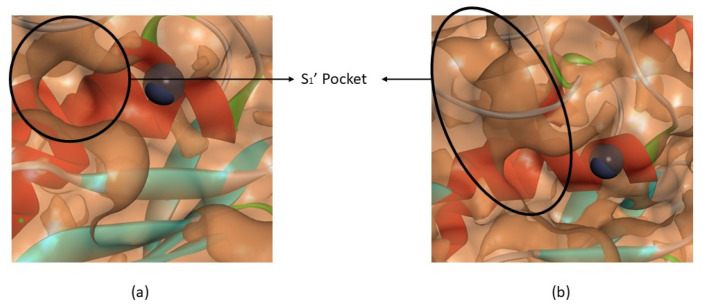
(**a**) Active site of MMP-1. (**b**) Active site of MMP-13. The S_1′_ pocket of MMP-1 is shallow and the S_1′_ pocket of MMP-13 is deep.

**Figure 3 biomedicines-09-00366-f003:**
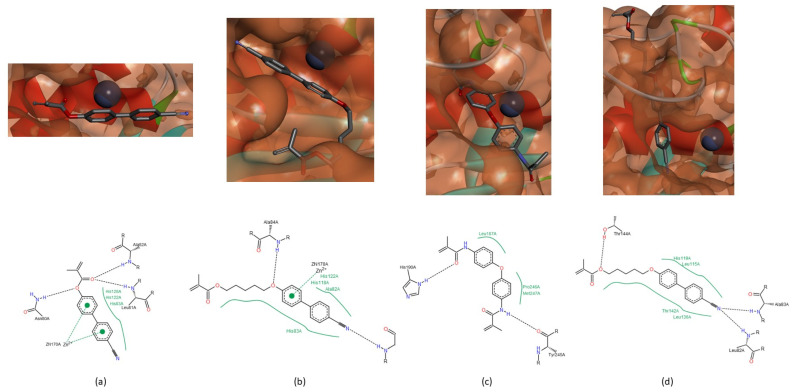
Location of compounds A, B and C in the active center of MMPs. (**a**) Compound A in the active site of MMP-1. The catalytic zinc ion can establish two interactions with the two aromatic rings, simultaneously. (**b**) Compound B in the active site of MMP-1. The catalytic zinc ion establishes an interaction with an aromatic ring. (**c**) Compound C in the active site of MMP-9. Compound C partially through the S_1′_ pocket. (**d**) Compound B in the active site of MMP-13 and the molecule totally through the S_1′_ pocket.

**Figure 4 biomedicines-09-00366-f004:**
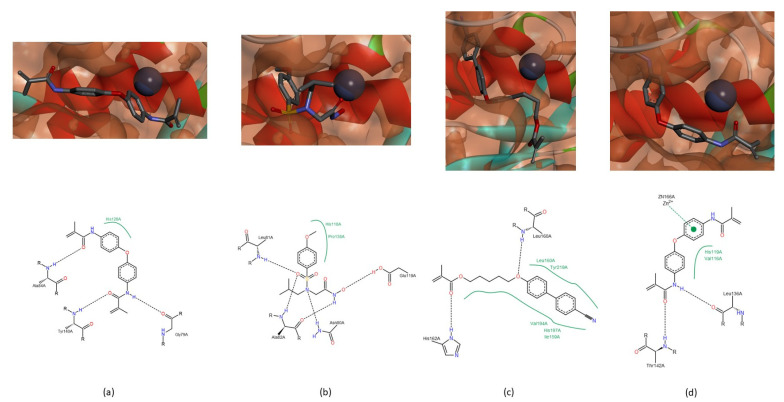
Interactions between MMP-1, -8 and -13 and the NNGH inhibitor and the compounds B and C. (**a**) The residue Gly_79_ and Tyr_140_, of MMP-1, establish a hydrogen bond with the NH group and the oxygen atom, respectively, of methylacrylamide function of compound C. (**b**) The Asn_80_ of MMP-1 establishes a hydrogen bond with the tertiary amine of NNGH inhibitor. (**c**) The His_162_ of MMP-8 interacts with the oxygen atom of the carbonyl group of methacrylate of the compound B. (**d**) In MMP-13, the residue Thr_142_ establishes a hydrogen bond with the oxygen atom of methylacrylamide of compound C.

**Figure 5 biomedicines-09-00366-f005:**
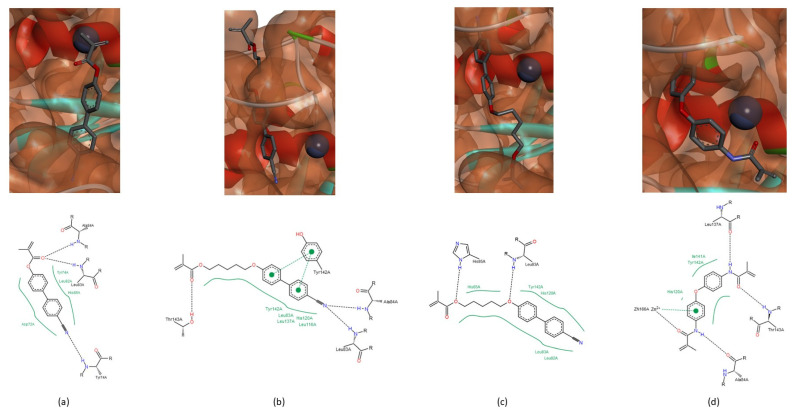
Interactions between compounds A, B and C and the MMP-2. (**a**) Compound A. The oxygen atom of the carbonyl group of methacrylate establishes hydrogen bonds with Leu_83_ and Ala_84_, simultaneously. (**b**) Compound B. The CN group establishes hydrogen bonds with Leu_83_ and Ala_84_, simultaneously. The residue Tyr_142_ interacts with the oxygen atom of the carbonyl group of methacrylate and the residue Thr_143_ interacts with the biphenyl group. (**c**) Compound B. The Leu_83_ and His_85_ establish a hydrogen bond with the oxygen atom of the methacrylate group and with the oxygen atom adjacent to the biphenyl group. (**d**) Compound C. The Ala_84_, Leu_137_ and Thr_143_ establish a hydrogen bond with the methylacrylamide group.

**Figure 6 biomedicines-09-00366-f006:**
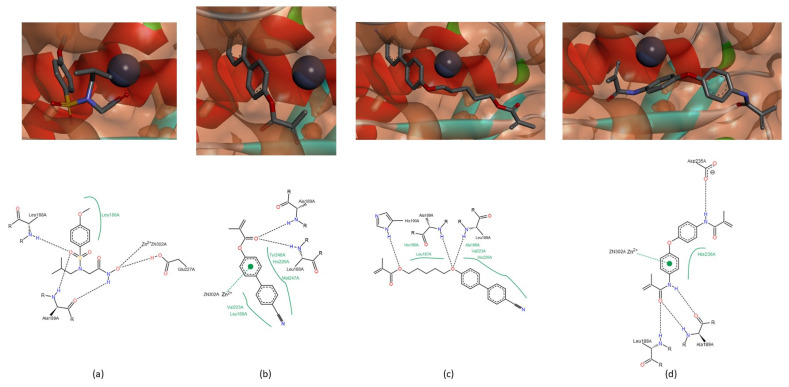
Interactions between the NNGH inhibitor and the compounds A, B and C and the MMP-9. (**a**) NNGH inhibitor. Leu_188_ and Ala_189_ interact with the oxygen atom of the sulfonyl group and the NH group of hydroxamic acid. (**b**) Compound A. Leu_188_ and Ala_189_ establish a hydrogen bond with the oxygen atom of the carbonyl group, simultaneously. (**c**) Compound B. Interactions between Leu_188_, Ala_189_ and His_190_ and the oxygen atom of methacrylate and adjacent to the biphenyl group, respectively. (**d**) Compound C. The methylacrylamide can establish a hydrogen bond with Leu_188_, Ala_189_ and the residue Asp_235_.

**Figure 7 biomedicines-09-00366-f007:**
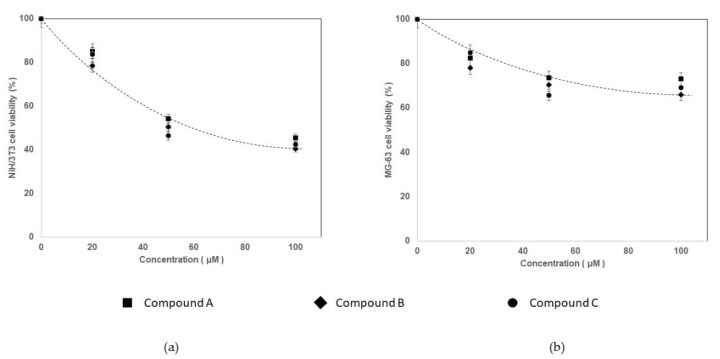
NIH/3T3 (**a**) and MG-63 (**b**) cell viability upon exposure to compounds A (4′-cyano-[1,1′-bihenyl]-4-yl methacrylate), B (5-((4′-cyano-[1,1′-biphenyl]-4-yl)oxy)pentyl methacrylate) and C (N,N′-(oxybis(4,1-phenylene))bis(2-methylacrylamide).

**Table 1 biomedicines-09-00366-t001:** MMP information obtained from the Protein Data Bank (PDB).

MMP	PDB ID	Total Structure Weight (kDa)	Method	Resolution (Å)	R-Factor (%)
1	2tcl	19.63	X-ray diffraction	2.20	16.2
2	1HOV	19.28	Solution NMR	−	−
8	1BZS	19.22	X-ray diffraction	1.70	19.2
9	4XCT	18.97	X-ray diffraction	1.30	17.0
13	1FM1	19.21	Solution NMR	−	−

**Table 2 biomedicines-09-00366-t002:** Physicochemical and ADMET properties of the NNGH inhibitor and the compounds A, B and C.

Property	NNGH	Compound A	Compound B	Compound C
Molecular weight (MW)	316.375	263.295	349.428	336.389
H-bonds acceptor	6	3	4	2
H-bonds donor	2	0	0	2
Log P	1.472	3.828	5.089	2.943
Log D	−0.075	3.828	5.089	2.943
Topological polar surface area (TPSA)	95.94	50.09	59.32	67.43
BBB category	−	+	+	−
BBB log[(brain): (blood)]	0.703	−0.158	−0.15	−0.492
P-gp category	No	No	No	Yes

**Table 3 biomedicines-09-00366-t003:** Interactions between the residues Leu, Ala and Ala of the MMPs-1, -8 and -13 and the NNGH inhibitor and the compounds A, B and C.

Residue	NNGH	Compound A	Compound B	Compound C
Leu_81_ (MMP-1); Leu_160_ (MMP-8); Leu_82_ (MMP-13)	All collagenases	All collagenases	MMP-8 MMP-13	MMP-8
Ala_82_ (MMP-1); Ala_161_ (MMP-8); Ala_83_ (MMP-13)	All collagenases	MMP-1	MMP-8 MMP-13	MMP-8 MMP-13
Ala_84_ (MMP-1); Ala_163_ (MMP-8); Ala_85_ (MMP-13)	No interact	MMP-13	MMP-1	All collagenases

**Table 4 biomedicines-09-00366-t004:** Percentage of mean (M) and standard variation (SD) of cell viability for the two type of cells with the several inhibitor compounds in different concentrations.

Groups		Cell Viability (%)			
**Inhibitor** **Compound**	Inhibitor Concentration	NIH/3T3		MG-63	
		M	SD	M	SD
A	20 μM	85.1	0.71	82.6	2.28
	50 µM	54.2	4.47	73.6	0.78
	100 µM	45.5	5.30	73.0	5.40
B	20 µM	78.6	1.41	78.1	1.41
	50 µM	50.5	4.42	70.3	2.03
	100 µM	40.5	1.60	65.9	0.82
C	20 µM	83.7	1.41	85.0	0.36
	50 µM	46.4	4.22	65.8	2.94
	100 µM	42.4	8.28	69.2	9.24

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
