# Peer review of "Polymerizable Matrix Metalloproteinases’ Inhibitors with Potential Application for Dental Restorations"

_biomedicines, 2021, doi:10.3390/biomedicines9040366_

Round 1

Reviewer 1 Report

Comments to the Authors

  1. The current study is mainly focused on understanding three different compounds as MMP inhibitors through computational biology & bioinformatics. However the study is more hypothetical with evidences from previous reports. The authors haven’t performed any experiments to prove and substantiate the computational observations, wherever possible. Secondly, how the current system is beneficial in dental restoration, as no experimentation has been performed. It’s merely the material-material interaction which has been explored which may be suggestive of a future application in dentistry. Hence I would suggest the authors can do some in vitro experimentation to prove the usefulness of the selected compounds as MMP inhibitors.
  2. The title must be modified. Computational part can be included whereas dental restoration can be removed from the title.
  3. Line no.77 reads as “Two fibroblastic cell lines”, one is NIH 3T3. Which is the other fibroblastic cell line used? MG 63, human osteosarcoma cell line, has fibroblast-like morphology, but I don’t think it’s a fibroblastic cell line. Hence either clarify with appropriate references or make necessary changes.
  4. Give expansions for DMEM and EMEM.
  5. The procurement source must be mentioned in brackets for each and every material like medium, serum, antibiotic solution etc.
  6. It is essential to mention at which passage the cells were used for the cytotoxicity/cellular experiments. Also the seeding density of the cells for a 96 well plate.
  7. Rather than just mentioning the compound A, B and C in the figure legend, it would be appropriate to include the compound names under sections 2.2 and 2.3. In future, each time it’s not possible for the reader to scroll to the top or to look into the reference in order to find out the names of the compounds. The authors must look into such flaws and make necessary changes throughout the manuscript. The usage of codes like A, B & C can be restricted to figures & tables alone.
  8. What is the solvent medium for each of the compound as mentioned in lines 84-86?
  9. In figure 7, change the word cell viability to cytotoxicity
  10. Look for grammatical errors and spelling mistakes throughout
  11. What’s the relevance of dental resins & restorations in the title as no study has been performed to prove its efficacy.

Author Response

to whom it may concern

thanks for the comments and corrections, all the point have been addressed ( see below)

Jorge Caldeira

1 The comments were appropriate in regard to the need of in vitro experimentation. We do intend to publish the results of the ongoing work. A lot of experimental challenges were overcome regarding solubility, photopolymerization and biological activity that need to be addressed in a separate publication. Nevertheless, we haven’t emphasized enough in the manuscript that the computational studies reported in this paper enabled to select these three compounds from several hundreds of other proposals that were less promising in terms affinity towards the active center.

A sentence to give that context was included now in the manuscript. but the reference in vitro experimentation was just restricted to other crucial factor toxicity that is addressed in this work.

2 The tile was changed

3 Clarification regarding this point was included in the manuscript.

4 Definition of DMEM and EMEM included.

5 Chemical and reagents suppliers included

6 Description of A B C compounds included thru the text

(A- 4’-cyano-[1,1’-bihenyl]-4-yl methacrylate; B- 5-((4’-cyano-[1,1’-biphenyl]-4-yl)oxy)pentyl methacrylate; C-N,N’-(oxybis(4,1-phenylene))bis(2-methylacrylamide))

7 the medium added was the cell culture media, (now included in the text)

8 We think we should maintain cell viability since it is the direct experimental observation.

9 Corrections were made and include in the manuscript

10 Change was made (see above).

11 Clarification of the relevance included in the text

Reviewer 2 Report

Your manuscript was read with interest. Please make sure your conclusions are supported by your findings.

Do not forget to report the limitations of your study and the potential for improvement in future studies.

Author Response

to whom it may concern

thanks for the comments and corrections, all the points have been addressed in the final text.

Jorge Caldeira

Reviewer 3 Report

General comments

This paper includes a novel idea to improve the resin matrix of resin composite for long term degradation. However, there are some questionable and insufficient issues to be corrected as follows:

Title page

Please change “Barahona1, Jorge Caldeira 1,2 and *” to “Barahona1 and Jorge Caldeira 1,2 *

Introduction section

There was no clear description about the aim of this study. The null hypothesis is also necessary.

line 24

“The matrix metalloproteinases (MMP)” should be corrected as “The matrix metalloproteinases (MMPs).

The authors should uniform the description of “NIH 3T3” and “MG-3” through the whole. The descriptions of “3T3”, “NIH3T3” and “MG3” were included.

Materials and Methods section:

2.2. Synthesis of A, B and C Compounds (line 74-75)

The authors described “Synthesis was performed according to the reference [42]”; however, methodology for the synthesis of the compounds is unclear. The authors should explain the outline of the methodology to obtain understanding from readers.

line 88: CO2 should be changed to CO2.

2.3. Cytotoxicity study using MTT assay (line 76-97)

Why did the authors use two kinds of fibroblast cell (NIH 3T3 and MG-63) for MTT assay?

2.4. Statistical Analysis (line 98-103)

In this study, two-way ANOVA was used to analyze the effect of cell types (main factor, two levels) and experimental groups (main factor, nine levels) on the cytotoxicity of the new products. However, the cell type seems to be unsuitable as a main factor. It could be more important to prove the effect of the kinds and the concentrations of the compound on the cytotoxicity for each fibroblast. Please try two-way ANOVA designed by main factor A (the kinds of the compound, three levels of A, B and C) and main factor B (the concentrations of the compounds, three levels of 20, 50 and 100µM) for each fibroblast cell.

Results and Discussion section

Please rewrite the results of cytotoxicity test according to the results statistically analyzed using the two-way ANOVA recommended.

Conclusions section

line 299-303: the results of the preliminary test should not be described in conclusion. The author should transfer these sentences to results and discussion section, or eliminate.

Author Response

to whom it may concern

thanks for the comments and corrections, all the point have been addressed ( see below)

Jorge Caldeira

i) Names were corrected

ii) Aim of the study more clearly stated and logical implications addressed.

iii) Correction included in the manuscript.

iv) Description of 3T3”, “NIH3T3” and “MG3” now included

v) The organic chemistry methodology is now outlined in the manuscript

vi) Changed in the paper

vii) Two cell lines were used for further confidence in the results

viii) correction of the statistical  procedure 

ix) clarification of the statistical methodology include in the text

x) Preliminary results moved to result section according to suggestion